# Uncovering the Hidden Data Costs of Mobile YouTube Video Ads

## ABSTRACT

Popular video streaming platforms attract a large number of global marketers who use the platform to advertise their services. While benefiting platforms and advertisers, users are burdened with the costs of advertisements. First, users indirectly pay for these ads, not just in terms of the time they invest but also through the sharing of personal information. Such exchanges might be justifiable if they resulted in a free and seamless Internet experience. However, the reality is that these ads also consume a substantial amount of data, translating into direct financial costs for the users. This issue becomes even more pronounced in developing countries, where the cost of mobile broadband in these countries is disproportionately high relative to average income levels. In this paper, we perform the first independent and empirical analysis of the data costs of mobile video ads on YouTube, the most popular video platform, from the users' perspective. To do so, we collect and analyze a data set of over 46,000 YouTube video ads. We find that streaming video ads have multiple *latent* and *avoidable* sources of data wastage, which can lead to excessive data consumption by users. We also conduct an affordability analysis to quantify the cumulative impact of data wastages and unveil country-specific data costs associated with these losses. Our findings highlight the need for video platform providers like YouTube to reduce data wastage associated with ads to make their services more affordable and inclusive.

## 1 INTRODUCTION

Over 5 billion Internet users worldwide watch billions of hours of online video every day—with three out of five video views coming from mobile devices [8]. The growing popularity of online video platforms has drawn many marketers. Marketers use these platforms to advertise their services, whereas video platform providers monetize their services via personalized advertisements; e.g., YouTube had a global ad revenue of USD 29.2 billion in 2022 and had drawn more than 50% of the global marketers [15].

While advertisements have become a ubiquitous presence in our online experiences, the hidden costs borne by users are often overlooked. First, users indirectly pay for these ads, not just in terms of the time they invest but also through the sharing of personal information. Such exchanges might be justifiable if they resulted in a free and seamless Internet experience. However, the reality is that these ads also consume a substantial amount of data, translating into direct financial costs for the users. This issue becomes even more pronounced in developing countries, where the primary mode of Internet access is through mobile devices [18]. Compared to developed countries, the cost of mobile broadband in these countries is disproportionately high relative to average income levels [13]. This disparity means that for a vast majority, every megabyte of data is a precious resource. A study by the World Bank across 11 emerging countries found that nearly half of the respondents had difficulty paying for their mobile data usage and 42% had to impose self-restrictions on their data usage, which inadvertently limits

their access to the vast resources and opportunities the Internet offers [12].

The gravity of this issue is further underscored by the fact that, as of 2021, 95 countries did not meet the UN Broadband Commission's affordability benchmark for broadband services. This target, set at 2% (or less) of the monthly Gross National Income (GNI) per capita,[1] is indicative of the digital divide that continues to impact the global community [9, 10]. Thus, it is crucial to recognize and address the unintended economic burdens ads place on users, especially in regions where access to the Internet is not a given but a luxury.

In this paper, we perform the first independent and empirical analysis of the data costs of mobile video ads on YouTube from the users' perspective. We find that streaming video ads have multiple *latent* and *avoidable* sources of data wastage, which can lead to excessive data consumption by users. Our findings highlight the need for video platform providers like YouTube to reduce data wastage associated with ads to make their services more affordable and inclusive.

We focus our study on YouTube as it is the most popular online video platform with approximately 2.1 billion users worldwide [15]. We devise a methodology to systematically collect, clean and analyze a large video dataset of a cumulative total of 17,600 YouTube videos (referred to as main-videos) and over 46,000 video ads. The aggregate duration of all videos (including ads) in our dataset amounts to 8,225 hours. To understand and analyze cross-country variations, we conduct this study across eight sampled countries, including four developing and four developed countries (for our sampling strategy refer to §2.2).

Using the streaming data collected for YouTube videos, we first analyse the proportion of overall data consumption attributed to video ads. We then perform a deeper analysis based on client's video buffer states and ad placement policies to uncover the hidden costs arising from instances when video data is wasted as a direct consequence of these ads. Below, we highlight the key insights from our measurement study.

- **Video Ad Data Consumption:** We observe that video ads account for an average of 13.2% of total video bytes in our dataset (which included both developed and developing countries). This percentage considers scenarios where users *always* skip the skippable ad portions.[2] However, when users *do not* skip ads, the contribution of video ads increases to an average of 28%.
- **Data Wastage with Skippable Ads:** Our analysis found that when a user skips the skippable portion of a YouTube in-stream ad, they have already downloaded a significant amount of data (video chunks) for that portion of the ad. This data is wasted when a user has already clicked the skip button and will not view the skippable portion of the ad. This happens because ads are aggressively buffered. In developing regions, 31% of skippable ads

---

[1] This is for a 2 GB data-only mobile broadband plan. Recently, the Alliance for Affordable Internet has revised these affordability targets and encourage governments across the world to set targets such that the cost of 5GB of broadband, both mobile and fixed, should not be more than 2% of the average monthly income by 2026 [11].

[2] This represents a lower bound on the data cost of video ads.

were mostly downloaded (between 80-100%) before users clicked the 'skip' button. In developed regions, this proportion was even higher at 52.6%, which is an increase of 1.7×. The implications of skip-loss are significant since most users skip ads [1] but are still being charged for content they did not watch.

- **Skippable vs Non-Skippable Ads:** Analysis of the placement of ads highlights how a large proportion of skippable ads are positioned at the beginning of the main-video while non-skippable ads are evenly distributed across the duration. Based on the position and frequency of both types of ads, we find that users are more likely to encounter a skippable ad, and hence experience data wastage due to skippable ads.

- **Data Wastage due to Mid-Roll Ads:** Our study finds that in-stream video ads that appear while a video is playing (also called *mid-roll ads*) cause buffered main-video chunks to be discarded and then re-downloaded when the ad finishes. We observe that on average, a user had to re-download approximately 4.2% of the main-video in developing regions, while in developed regions users had to re-download 5.8% of the main-video, which adds to the data cost. We also observe that mid-roll ads placed in the first half of the main-video lead to greater buffer losses and consequent re-buffering than those placed in the later half. These losses appear on *both* the mobile YouTube app and `m.youtube.com`.

- **Affordability of Video Access:** Our analysis shows that for developing countries in our dataset, video ads alone consume 9.2% of the baseline 2 GB mobile internet data plan (under the assumption that all skippable ads are skipped). This has significant implications for users in developing countries, where mobile data plans cost a significant fraction of the per-capita income [6].

- **Potential Solution and Impact:** We discuss a potential solution to minimize video buffer loss due to mid-roll ads. Our analysis shows that the proposed solution can reduce the average data consumed by 1.2×. This reduction allows a user to watch an additional 7 mins worth of video on a 2 GB data-plan. We also show that an optimal solution that does not incur any excess buffer loss allows a user to watch an additional 38 mins of video on a 2 GB data-plan.

- **Validation Across Platforms:** By default, we conduct experiments using YouTube Mobile Web. In addition, we conduct validation experiments using YouTube mobile app on multiple Android smartphones. We observe that Android YouTube app also experiences video buffer losses due to skippable ads and mid-roll ads (§3.5).

These findings have significant implications for various stakeholders in the video-streaming ecosystem. Our study highlights opportunities for video-streaming platform providers like YouTube to improve their services by making video accesses more affordable for mobile broadband users. For example, they can develop more intelligent media player applications that can differentiate between the type of video being rendered (e.g., ad-video vs. main-video) to reduce video buffer wastage. They can also adapt video buffering to make video access more affordable and inclusive. Furthermore, our study suggests that extending ad systems to incorporate video loss considerations in deciding ad placements can improve video access affordability. By implementing these measures, video-streaming platform providers can reduce data costs for users and increase

video accesses to their platforms. Moreover, our study has identified hidden data costs that users incur while watching videos on YouTube and it can help users make informed decisions about their video streaming habits.

Altogether, we make the following *key* contributions.

- We devise an experimental methodology for systematically collecting and cleaning a large corpus of video ad data. Our methodology allows for automatically crawling videos and extracting ad video data. We stream 17,600 videos, over 46,600 ad videos, totalling approximately 8,225 hours. We share an anonymized link to our code and data for the community *here*.

- We analyze video ad data consumption as well as conduct an in-depth video buffer analysis. Our study reveals multiple hidden data costs of viewing videos and highlights the need for video platform providers to reduce data wastage associated with ads.

- We conduct a systematic affordability analysis to quantify the affordability of video streaming with in-stream ads across all countries in our dataset. Our analysis points to the need for video platform providers to make video accesses more inclusive and affordable.

- We discuss the implications of our study on different stakeholders. In particular, we suggest solutions that can help reduce data wastage due to pre-emptive downloading of ad video data.

The remainder of this paper is organized as follows. First, we provide an overview of YouTube ads and describe our measurement methodology (§2). We then analyze video ad data (§3). We quantify the affordability of video streaming with in-stream ads across eight countries in (§4). This is followed by a discussion on the implications of our study (§5). We discuss related work in (§6) and finally conclude (§7).

## 2 MEASUREMENT METHODOLOGY

In this section, we provide an overview of video ads on YouTube and detail the methodology used to construct the YouTube dataset for our study.

### 2.1 Overview of YouTube Ads

YouTube provides marketers with different ad formats, including video and non-video options. This paper focuses on video ads that appear before, during, or after a YouTube video within the player (in-stream ads) to analyze the data cost of ads and particularly look into video buffer states. These ads are categorized into two types based on their format: *skippable ads* and *non-skippable ads*. Skippable ads allow viewers to skip them after a designated *time-to-skip* duration, while non-skippable ads must be watched entirely. Furthermore, ads can be classified based on their placement (i.e., when an ad appears within the video stream): (1) *pre-roll* ads appear before the video starts, (2) *mid-roll* ads appear during the video playback and (3) *post-roll* ads appear after the video has finished streaming. Consecutive ads shown back-to-back are referred to as *double ads*.

### 2.2 Sampling Methodology

Our study focuses on a sample of four developing and four developed countries, as categorized by the UN Human Development Report 2021-22 [19]. We included these countries in our dataset to

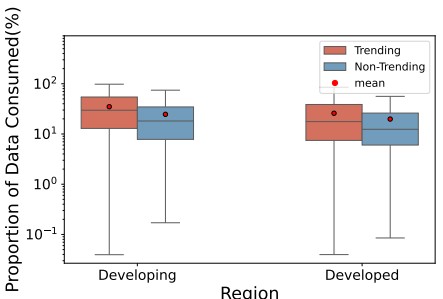 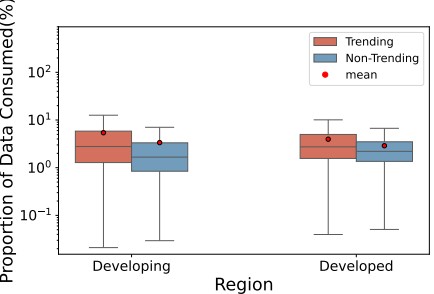

**Figure 1: Proportion of data consumed by ads considering only watch time. Ads watched entirely (left) and skippable ads skipped (right).**

ensure the representation of a significant proportion of YouTube users. We selected countries based on their ranking among the top 20 countries in terms of YouTube audience size [7]. Out of these 20 countries, we chose a convenience sample of four developing countries (Indonesia, Brazil, Mexico, and Pakistan) and four developed countries (USA, Canada, Germany, and Japan). This selection covers a total of 862.9 million YouTube users. Our dataset comprises 17,600 YouTube videos, referred to as the *main-video* in this paper. For each country, there are two categories of main-videos collected:

- **Trending Videos:** Videos on the YouTube Trending page are categorized based on their high view count and *temperature*.[3] Trending videos represent the most watched videos in a country.
- **Non-Trending Videos:** Videos present on the YouTube homepage, but not on the Trending page.

## 2.3 Video Streaming and Data Collection

For each country, the data collection process was divided into two phases. In Phase 1, a Selenium-based web scraper (henceforth, $S_1$) was used to scrape the URLs of trending and non-trending main-videos. In Phase 2, another Selenium script (henceforth, $S_2$) was used to stream the URLs and collect data of interest. Both phases of data collection were repeated daily over a period of two months from 2023-03-01 to 2023-05-01, resulting in a dataset consisting of 1,100 trending and 1,100 non-trending main-videos for each country. When crawling the web scraper to collect the video URLs, we automated the pipeline using CRON [2] to ensure that the videos being streamed were scraped at the same local time-of-day. In total, our dataset comprises 17,600 main-videos and 46,613 video ads, with a combined duration of 8,225 hours.

**Video Streaming.** To replicate the network conditions for each country, $S_2$ was throttled to the Average Mobile Bandwidth (AMB) for that country, obtained from Open Signal [14]. It also ran in mobile emulation mode, emulating Nexus 5 Build/JOP40D, and spawned a new Google Chrome instance (version 111.0.5563.64) to stream the URLs on YouTube Mobile Web (`m.youtube.com`). It is important to note that using YouTube Mobile Web ensured that each streamed URL was provided with the same environment, Chrome version, and setup, eliminating variations caused by different mobile operating systems, YouTube application versions, and versions of the same mobile OS. This standardization was crucial for fair and *unbiased* cross-country comparisons, especially considering

the additional variance introduced by different device types, OS preferences, and mobile OS versions across countries. Additionally, all videos were streamed at the default resolution of 360p to ensure consistency across the dataset. Consequently, all ads were also streamed at the automatic 360p resolution chosen by the player. Lastly, $S_1$ and $S_2$ were run on 8 Ubuntu 22.04 LTS-powered machines, and each machine was connected to one of the 8 countries in our sample using a commercially available Virtual Private Network (VPN) service.

**Data Collection.** Within each main-video, ads were identified by monitoring changes to the HTML5 video player and relevant ad-data such as the type (skippable/ non-skippable), time-to-skip duration, and timestamp (i.e., the streamed duration of the main-video at which an ad appears) were collected using the HTML5 video player. Metadata for the videos, such as the video ID, resolution, and buffer (the additional prefetched seconds of video content), was obtained by enabling the YouTube 'Stats for Nerds' interface.

## 3 ANALYSIS AND RESULTS

In this section, we first investigate the data consumption of watching all video ads within a main-video. We then reveal two hidden buffer losses directly associated with YouTube ads and shed light on the data consumed by YouTube ads, which is significantly higher than the data associated with ads watched by the user. Moreover, we examine the impact of streaming resolution on the data consumed by ads. To quantify ad consumption, we express the data associated by ads *relative* to the total data consumed during the streaming of the main-video. This measure is referred to as the *ad data proportion.*

### 3.1 Ad Data Consumption based on Watch Time

We begin our analysis by examining the primary variable of interest: ad data proportion. In this section we consider a scenario where users are only charged for the content they watch, with no inclusion of any potential hidden costs associated with the ads. We consider both scenarios: one in which a skippable ad is skipped and another in which the user watches the entire duration of the skippable ad. The visualization of these results is presented in Figure 1.

As expected, we observe a significant decrease in the data consumed by ads when users choose to skip the skippable ad. This reduction can be substantial, with a maximum decrease of 10.8×

---

[3]The rate at which a video generates views [21].

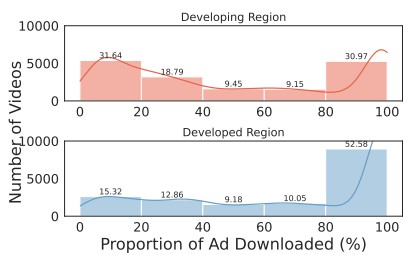

Figure 2: Proportion of skippable ad
downloaded at time-to-skip.

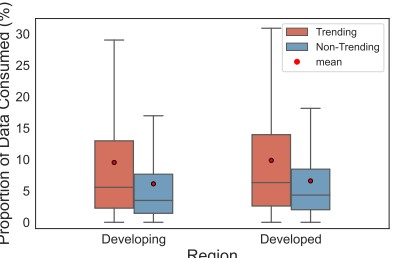

Figure 3: Proportion of data
consumed by skip-loss.

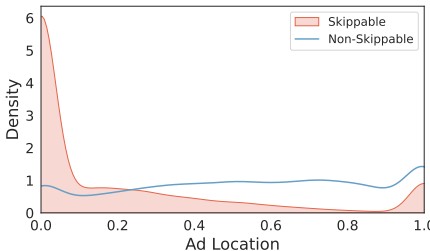

Figure 4: Skippable and non-skippable ad
location.

for non-trending videos in developing regions, and a minimum reduction of 5.6× for non-trending videos in developed regions.

**Ad Proportion on Trending vs Non-Trending Videos.** Our analysis reveals a notably higher ad data proportion associated with trending main-videos in comparison to non-trending main-videos. When users watch the complete skippable ad, ads on trending main-videos constitute a median of 22.9% of the data consumption, whereas non-trending videos in the same scenario account for a median of 15% of the data. Similarly when skippable ads are skipped, ads on trending videos result in a median of 2.7%, compared to 2% for non-trending videos. The higher proportions of ad data in trending videos are unexpected, especially given that our dataset indicates fewer ads (19,445) in trending main-videos compared to non-trending main-videos (27,168). However, we attribute this difference to the significantly shorter duration of trending main-videos (315 secs) compared to non-trending main-videos (1,081 secs). Hence, the ad data proportion is greater due to the increased number of ads per unit time of main-video.

**Differences in Ad Gaps.** Next, in our exploration of ad frequency and distribution, we introduce a novel metric termed the "Ad Gap," representing the time elapsed before an ad is displayed during a main-video. A smaller Ad Gap value signifies a higher frequency of ads shown on a main-video per unit of time. Our analysis uncovers that, on average, an ad appears after 4.6 mins on trending main-videos and every 6.1 mins on non-trending main-videos. Additionally, users in developing regions encounter an ad after around 4.9 mins of content streamed, while those in developed regions see an ad after approximately 6 mins. Based on these findings, we can conclude that when watching main-videos of equivalent durations, YouTube users are more likely to encounter more ads when watching trending videos than non-trending videos or when streaming main-videos in a developing region than a developed region within the scope of our dataset.

### 3.2 The Hidden Cost of Skippable Ads

In this section, we unveil a hidden cost linked to skippable ads, which has significant implications for the affordability of video streaming. Among the 46,613 video ads in our dataset, 74.6% are skippable ads, while the remaining 25.4% are non-skippable ads. Additionally, we examine ad placements to gain further insights into these ad formats.

**Skip Loss.** In an ideal scenario, users should *only* pay for the portion of the skippable ad they actually watch in terms of data

consumption costs. However, we observe the contrary. Excessive buffering of ads, beyond time to skip (5 secs), result in buffer loss when the user skips the ad. Figure 2 reveals the extent of this buffering at time-to-skip for an equal number (17,000) of skippable ads from the developed and the developing region. Within the first 5 secs, 80%-100% of the ad is downloaded for a surprising 52.6% of all skippable ads in developed regions and 31% in developing regions. Once a user skips the ad, all the excess buffer downloaded beyond the skippable time, for the unwatched content goes to waste. We term this avoidable buffering and subsequent loss as **skip-loss**. The implications of skip-loss are significant since most users skip ads [1] but are still being charged for content they did not watch.

**Impact of Skip-Loss.** Next, we quantify the impact of skip-loss. We find that skip-loss consumes 5.9% of the total data consumption of the main-video for trending main-videos and 3.9% for non-trending main-videos; see Figure 3. Similarly, for the developed and developing regions, the proportion of data consumed by skip-loss is 5.1% and 4.3% respectively. The mean difference in skip-loss between trending and non-trending groups (0.8%) was statistically significant ($p < .001$). Moreover, the mean difference in skip-loss between developing and developed regions was also statistically significant at the 5 percent level ($p = .0164$). For this analysis, the size of the *skippable* portion of the ad (to calculate ad data proportion) is modeled as follows: (1) the size of the portion of the ad that is watched by the user until time-to-skip and (2) the size of additional ad content that is downloaded beyond the time-to-skip and stored in the buffer. It is interesting to note that the proportion of data consumed just by all skip-losses is higher than the proportion of data consumed by the watched content of all ads if users always skip at time-to-skip (highlighted in Figure 1).

**Skippable Ad Location.** Lastly, we make an interesting observation about the placement of different formats of video ads within a main-video. To analyze this, we define "Ad Location" as the ratio of the ad timestamp to the duration of the main-video. Figure 4 shows the kernel density plot for Ad Location of skippable and non-skippable ads in the dataset. We observe that a large proportion of skippable ads are positioned at the beginning of the main-video with their density progressively declining throughout the remainder of the video except at the very end. Conversely, non-skippable ads have a relatively even distribution across the duration of the main-video. More specifically, 69.1% of the skippable ads lie between the Ad Location 0.0 and 0.2, whereas only 17.3% of non-skippable ads lie within this Ad Location. Given the high concentration of

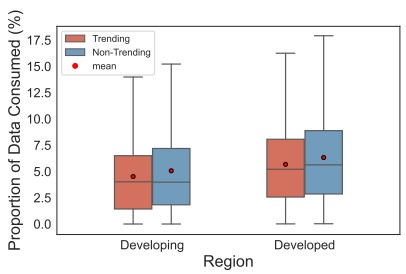

Figure 5: Percentage of main-video lost due to mid-roll ads.

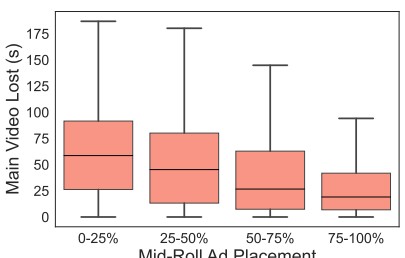

Figure 6: Proportion of data consumed by mid-roll buffer loss.

Figure 7: Main-video lost due to mid-roll ads categorized by placement.

skippable ads at the start of the video, we can presume that even users who decide to quit before watching the entire main-video encounter a greater number skippable ads than non skippable ads.

In conclusion, we gather two key takeaways from the analysis of skippable ads. First, excessive buffering and pre-fetching of the ad content beyond time-to-skip leads to significant data loss when the user decides to skip the ad. Secondly, given the ad-counts and location, users are more likely to encounter skippable ads compared to non-skippable ads, resulting in a higher frequency of incurring the overhead cost associated with skip-loss.

## 3.3 The Hidden Cost of Mid-Roll Ads

In this section, we present the analysis of mid-roll ads. We highlight an avoidable buffer loss associated with mid-roll ads and quantify its impact on the ad data proportion. Additionally, we examine the influence of mid-roll ad placement on this buffer loss.

**Mid-Roll Buffer Loss.** Analysis of the main-video buffer states unveils another hidden and unnecessary cost borne by YouTube users. We examine the main-video buffer right before and after a mid-roll ad appears and observe how any buffered content drops to zero as soon as a mid-roll ad begins playing. This implies that the additional seconds of the main-video content that the user has yet to watch, but has already paid for in terms of data costs, are effectively wasted when the mid-roll ad is encountered. Consequently, once the ad concludes, the buffer needs to be re-downloaded, leading to redundant data consumption. We term the loss of the main-video buffer, and the consequent re-downloading, as **mid-roll buffer loss**.

To quantify mid-roll buffer loss, in Figure 5 we plot the percentage of the main-video lost due to this issue. This percentage represents the cumulative mid-roll buffer losses on a main-video, relative to its duration. On average, in developing regions, 4.2% of the main-video is lost due to mid-roll ads, while in developed regions, this number increases to 5.8% of the main-video. The observed difference (1.6%) is statistically significant at conventional significance levels ($p < .001$). Similarly, 4.9% of main-video is lost due to mid-roll ads on trending videos, and 5.1% is lost for non-trending videos. The difference between trending and non-trending groups is also statistically significant ($p < .001$) . The higher proportion for developed regions can be attributed to the relatively greater bandwidth availability in these countries. This results in a larger buffer state of prefetched video content, which ultimately gets lost when a mid-roll ad plays.

**Impact of Mid-Roll Buffer Loss.** To analyze the impact of mid-roll buffer loss, we calculate its ad data proportion in Figure 6. We observe that in the developed region, mid-roll loss accounts for 5.5% of the total data consumption for a main-video, while in the developing region, it contributes to 4%. Similarly, mid-roll buffer loss for trending videos contributes approximately 4.7% to the total data consumption of the main-video, while for non-trending videos, it amounts to around 4.9%. The difference between the proportion of data consumption for trending and non-trending videos is statistically significant at conventional levels ($p < .001$).

**Mid-Roll Ad Placement.** We also examine the relationship between the placement of mid-roll ads and the resulting main-video buffer loss. We categorize mid-roll ads based on their placement and highlight the main-video buffer lost for each category of mid-roll ads in Figure 7. Our analysis reveals that ads positioned in the first half of the main-video result in a significantly higher mid-roll buffer loss compared to those appearing in the second half. Specifically, mid-roll ads in the first quarter result in a loss and subsequent re-downloading of approximately 71 secs of the main-video content, while ads in the second quarter contribute to a loss of about 58 secs. The observed difference was statistically significant ($p < .001$). For the third and fourth quarters, the loss decreases further to 43 secs and 34 secs, respectively. The difference between the third and fourth quarter was also statistically significant ($p < .001$). Considering that an equal number of mid-roll ads are randomly sampled for each placement category, the variation in main-video buffer loss can be attributed to the extent of buffering of video chunks in each placement category. These findings highlight the importance of considering ad placement in monetization policies to minimize unnecessary buffer loss and data wastage.

In summary, we quantified mid-roll buffer loss in this section. We also observed variations in the proportion of mid-roll buffer loss based on their placement within the main-video. Specifically, in-stream ads in the first half of the main-video result in higher buffer loss and data wastage compared to those appearing in the second half.

## 3.4 Impact of the Hidden Losses

Finally, we quantify the ad data proportion, taking into account not only the data consumed when streaming all ads (as in Figure 1) but also the data consumed by skip-loss and mid-roll buffer loss. This proportion is represented in Figure 8. We assume that all skippable ads are skipped at the 5 sec time-to-skip instance, hence our results

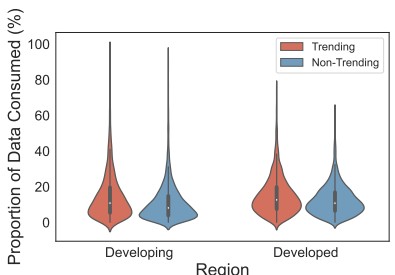

Figure 8: Proportion of data consumed by ads considering hidden losses.

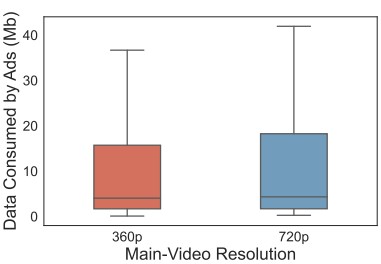

Figure 9: Data consumed by ads across main-video resolutions.

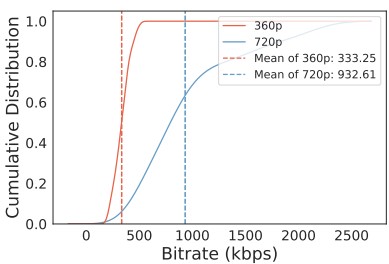

Figure 10: Distribution of average video bitrates across main-video Resolutions.

represent a conservative estimate for the ad data consumption. Comparing Figure 8 to Figure 1, we observe that excessive and unnecessary buffer losses, which are directly associated with ads, account for a significant portion of the data consumed by ads in each main-video. When taking into account these losses, ad data proportion for trending videos increases from 2.7% to 11.9%, while for non-trending videos it increases from 2.0% to 9.6%. Similarly, data consumption for ads in developing regions increases from 2.1% to 9.4%, and data consumption for ads in developed regions increases from 2.4% to 11.7%. On average, we observe a surprising increase of 4.7× in the ad data proportion if we take into account unnecessary buffer losses. Therefore, by uncovering the hidden buffer losses and their additional data costs, we provide a more comprehensive understanding of the true impact of video ads on users' data consumption.

## 3.5   Hidden Buffer Losses Across Platforms

While m.youtube.com provides an environment which is independent of platform and OS differences, we extend our results to the YouTube mobile app (version 18.39) on Android (across editions 11 and 13) in an effort to validate our m.youtube.com findings across different platforms. We used Android phones because of their popularity; over 3 billion active devices use Android worldwide [4].

We conduct an analysis of 50 mid-roll ads and 50 skippable ads. Data for mid-roll and skippable ads is collected by manually screen recording the video playback for each URL and retaining ad instances within each video. The results reveal that the Android YouTube app also experiences *both* skip-loss and mid-roll loss. Thus, the mid-roll buffer loss and skip-loss is a concern not just for m.youtube.com streaming, but also for the Android YouTube app. We leave a more detailed analysis for future work.

## 3.6   Comparison across Video Resolutions

In this section, we examine the influence of streaming resolution and video content quality on data consumption. We create two datasets that comprise streaming data collected under identical network conditions and streaming environments for the same ~500 trending main-videos in Pakistan. In dataset 1, the main-videos are streamed at 360p, while in dataset 2, the main-videos are streamed at 720p. The ad resolution stays consistent at 360p. Our objective in this analysis is to uncover any differences in data consumption patterns by video ads (if any) on YouTube across different streaming resolutions. We compute the data consumed by video ads and the

accompanying hidden buffer losses for each main-video in the two datasets and observed that in the 360p dataset, the average data consumed by ads and hidden losses per streamed main-video amounts to 13.5 MB. However, for videos streamed at 720p, there is a significant increase in ad data consumption, reaching 22.3 MB. The results are visualized in Figure 9.

Further analysis reveals that the disparity in the data consumed by ads between the two streaming resolutions can be attributed to the substantial difference in mid-roll buffer loss across the datasets. When main-videos are streamed at 360p, an average of 3.4 MB of data is lost to mid-roll buffer loss per video. Conversely, when streamed at 720p, the average data lost to mid-roll buffer loss increases to 10.1 MB. The variation in data lost to mid-roll buffer loss can be attributed to the higher bitrate requirements for videos streamed at higher resolutions. Typically, 720p videos have a higher bitrate compared to 360p videos. Consequently, heavier video data chunks are usually buffered and prefetched in the same unit of time, thereby increasing the likelihood of greater buffer loss due to mid-roll ad interruptions. We observed significant differences in the bitrates of the main-videos in our dataset. The mean bitrate for videos streamed at 360p was 333.3 kbps, while for videos streamed at 720p, it increased to 932.6 kbps, indicating a more than twofold increase as observed in Figure 10.

## 4   AFFORDABILITY ANALYSIS

In this section, we conduct a fine-grained country wise analysis of the cumulative buffer loss and its implications for cost and affordability. We explore cost through a data-plan lens which represents cost as a proportion of a base data plan. Next, we present a potential solution to minimize mid-roll buffer losses. We conclude with a what-if analysis which analyzes data consumption savings for different versions of video streaming.

## 4.1   The Real Cost of Ads

Section 3.4 quantified the proportion of data consumed by ads while taking into account all associated buffer losses. In this section, our objective is to understand the cost associated with ads and its country-wise implications for affordability. We adopt a data-plan perspective to quantify the cost which represents the percentage of data consumed for a fixed data-plan. To calculate data-plan cost, we assume a base 2 GB data plan subscription.

To understand the impact of ads and excessive buffer losses over a period of one month, we model a user that utilizes the entire

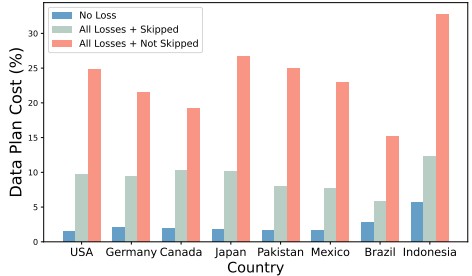

**Figure 11: Average monthly data plan costs for each country across three streaming cases.**

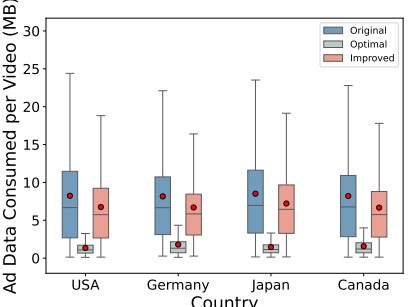 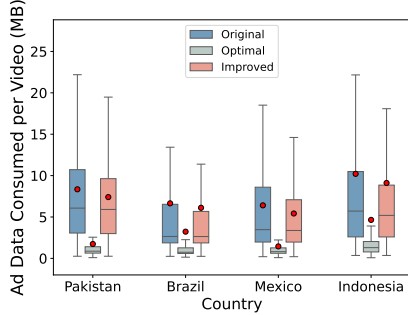

**Figure 12: Data consumed by ads across different types of video streams for the developed regions (left) and developing regions (right).**

**Table 1: Monthly data plan costs due to buffer loss.**

| Country | Average Buffer Lost (MB) | Data Plan Cost (%) |
|---|---|---|
| United States | 165.4 | 8.07 |
| Germany | 149.8 | 7.31 |
| Canada | 162.7 | 7.94 |
| Japan | 172.0 | 8.40 |
| Pakistan | 129.6 | 6.32 |
| Mexico | 121.2 | 5.92 |
| Brazil | 62.1 | 3.03 |
| Indonesia | 137.8 | 6.73 |

data-plan for video streaming on YouTube. In that case, we define video accesses for each country as the number of videos that can be streamed with a 2 GB data plan, considering the average data consumed when streaming a main-video (including the data cost of streaming the ads and the hidden losses) in each country. Next using video accesses, we compute the monthly data consumption associated with ads and the corresponding data-plan cost. Figure 11 represents this data-plan costs for different cases of ad data consumption. 'No Loss' corresponds to the scenario where there is no mid-roll buffer loss and skip-loss associated with the ads, and all skippable ads are skipped. Therefore, the only cost that is incurred is for watching the ad content. 'All Losses + Skipped' represents the case where ads incur both skip-loss and mid-roll buffer loss, and all skippable ads are skipped. 'All Losses + Not Skipped' refers to the case where skippable ads are not skipped. In this case, skip-loss is not incurred but mid-roll buffer is.

In Figure 11, we observe a significant difference in data-plan cost between the 'No Loss' and 'All Losses + Skipped' scenario. On average, the data-plan cost increases from 2.4% of the base 2 GB data for 'No Loss' to 9.2% for 'All Losses + Skipped.' This indicates that nearly 7% of the 2 GB data-plan is consumed by excessive buffer losses. Table 1 provides more detailed information on the data-plan costs associated with these losses. Furthermore, there is an even greater increase from the 'All Losses + Skipped' scenario to the 'All Losses + Not Skipped' scenario. Specifically, the data-plan cost rises to 23.5% from 9.2%. This increase is due to the higher data consumption when watching entire skippable ads and streaming the much longer skippable portions beyond the time-to-skip instance.

## 4.2 Alternative Video Streams

We now present a potential solution to mitigate the impact of mid-roll buffer loss on the cost of ads. Next, we analyze how ad data

consumption varies across different versions of video streaming on YouTube, including in the case of the proposed solution.

**A Simple Solution.** Building upon the insights from Section 3.3, we propose a straightforward solution to minimize buffer losses attributed to mid-roll ads. The root cause of the mid-roll buffer loss is the interruption of main-video by ads. The frequency of these interruptions directly impacts the extent of total mid-roll buffer loss for a main-video. Our proposal centers around reducing the total number of main-video interruptions by ads while maintaining the same number of ads. To achieve this, we utilize double ads. Note that double ads are already a part of the YouTube video streaming architecture but not all mid-roll ads are double ads.

We suggest replacing two single mid-roll ads with one double-ad, thereby reducing the frequency of interruptions of main-video streaming. When a double ad is encountered, the mid-roll buffer is lost as the first ad begins playing. As the second ad appears within the double-ad, there is no further loss of main-video buffer since it had already been emptied during the first ad's playback. However, if the second ad occurs at a distinct timestamp, encountering it will result in the loss of the main-video buffer once again. We evaluate the impact of this solution below as part of the *Improved Stream*. However, it is important to consider that the proposed solution solely aims to reduce the impact of excessive mid-roll buffer loss without taking into account any influence of double ads on user engagement and experience. Repeatedly encountering two consecutive ads during a video stream may negatively impact the user experience and cause early departure from videos due to increased waiting times before watching the desired content.

**What-If Analysis.** To understand the impact of the proposed solution, we conduct a what-if analysis across three different types of video streams: (1) Original stream: The current YouTube stream which suffers from mid-roll buffer loss and skip-loss, (2) Optimal stream: The counterfactual stream where there is no mid-roll buffer loss and no-skip loss, and (3) Improved stream: The stream incorporating double ads to minimize mid-roll buffer loss. Note that the improved stream will still incur all skip-losses.

Figure 12 depicts the distribution of total data (including the cost of ads and buffer losses) consumed by ads per main-video for both the developed and developing countries. We observe an improvement in data consumed by ads for both regions with the improved stream, as compared to the original stream. The average

buffer loss for the original stream in developing countries is 7.9 MB, which is improved to 7 MB with the improved stream. Similarly, the average data consumed by ads in developed countries for the original stream is 8.23 MB, which improves to 6.83 MB with the improved stream. For comparison, the optimal stream leads to an average data consumption of 1.5 MB in the developed countries, and 2.8 MB in the developing countries. The difference in the data consumed between developed and developing countries was statistically insignificant at the 5 percent level ($p = 0.071$). However, the difference between the improved stream and the optimal stream had a $p$-value less than .001. Finally, we highlight the video savings that can be realized across different streams. This provides a more context-specific perspective of the impact of excessive buffer losses. On average, the improved stream results in 422.5 secs (7 mins) more video playback per-month as compared to the original stream. Similarly, the optimal stream results in 2257.7 secs (38 mins) more video playback for a month, as compared to the original stream.

## 5 DISCUSSION

In order to make video streaming platforms such as YouTube more affordable and inclusive, it is important to address the issue of data costs associated with watching video ads. Our study highlights the need for platform providers to take action to reduce these costs. To this end, we propose a set of recommendations aimed at reducing the amount of data required to watch video ads and improving the overall user experience on the platform.

Firstly, our findings call for video-streaming platform providers to develop more intelligent and user-friendly media player applications that can differentiate between the type of video being rendered (e.g., ad video vs. main-video) and adapt video buffering to make the video accesses more affordable. There are multiple possible solutions: (i) client can stop main-video buffering before a mid-roll ad appears to prevent the loss of the already downloaded bytes due to ad interventions, (ii) a client-side player can maintain two separate video buffers (one for main-video, and the other for ad-video) so that the buffered main-video is not lost, and the user does not incur a startup delay after the mid-roll ad. However, this solution can increase the memory overheads on the client device, and (iii) the client player can be modified such that it does not pre-fetch skippable portion of ads to avoid skip-loss.

Our analysis also shows that ad placement has a significant impact on data wasted due to ads. YouTube and other video platform providers can incorporate data wastage constraints in their Ad Systems. For example, ad systems can make more strategic policies for the placement of mid-roll ads such that the buffer lost due to their intervention can be minimized. We quantify in the affordability analysis that one way to minimize the mid-roll buffer loss is to introduce more spread-out double ads instead of single ads. As discussed in the section 4.2, this refinement would cut down on the overall data consumption by 1.2×. Provided that skippable ads are more concentrated in the early quarters of the video as discussed in section 3.2 and they result in greater data loss than non-skippable ads due to the data consumption incurred by skip-loss, we call for dynamic skippable ad placement so they're evenly spread out across the entire video. Incorporating such affordability constraints in ad systems can help make video streaming inclusive.

## 6 RELATED WORK

In this section we highlight past works that are related to our study. We discuss related works in the following domains: data wastage, affordability, and YouTube ads.

**Data Wastage.** Recent efforts [20, 22, 23] revolve around data wastage in Adaptive Bitrate (ABR) streaming in the context of user-perceived QoE. While data wastage serves as a common motivator between these efforts and our work, there is an important distinction to be made between the nature of data wastage that is under investigation. Prior work focuses on main-video buffer lost due to the frequent occurrences of early departures by mobile users while streaming video. On the other hand, our work analyzes the cost of video ads and their associated hidden buffer losses. Any buffer loss resulting from user early departures would be in addition to the buffer losses we quantify. Moreover, our work emphasizes the data plan cost associated with buffer wastage and its implications for users with fixed cellular-data plans. To the best of our knowledge, our work is the first to explore such costs.

**Affordability.** While previous research has aimed to minimize data consumption in ABR video streaming and proposed cost-aware buffer management techniques [17], the motivation of these efforts has not been on understanding the affordability of the video streaming ecosystem. While some studies have touched upon web affordability [5, 16], the specific challenge of making video streaming more affordable has not been explored earlier. Given the drastic growth of video streaming over the internet, we believe such advances for video are equally important in the context of affordability and accessibility of the internet. Our work takes the first step in analyzing YouTube video streaming through an affordability lens and providing recommendations in making YouTube more affordable.

**YouTube Ads.** Prior work on YouTube ads primarily focused on user perception and engagement with video ads [1, 3]. However, unlike prior studies that mainly focus on the psychological perception of users towards YouTube video ads, our work represents a pioneering effort to conduct a comprehensive cost analysis from the user's perspective of these ads. Specifically, we evaluate the platform's ad strategies across various formats and placement policies to better understand the cumulative implications of these costs on the platform's affordability and inclusivity.

## 7 CONCLUSION

In this work, we conducted the first large-scale empirical analysis of YouTube with the goal of understanding the data costs of video ads through an affordability lens. We collect and analyse a dataset of over 17,000 videos and 46,000 ads across 8 countries. Our findings showed that on average 13-28% of the data is consumed by YouTube video ads, and surprisingly a significant amount of additional data is wasted because of (i) aggressive ad buffering of skippable ads and (ii) interplay between the main-video and mid-roll ad video media. We further conduct a country-specific affordability analysis, revealing the implications of buffer wastages and highlight potential monthly data savings for users if these hidden buffer costs are avoided. We discuss solutions and recommendations for video platform providers to reduce data wastage associated with video ads to improve video access, and make video streaming more affordable and inclusive.

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

# A ETHICS

This research paper adhered to strict ethical standards throughout the study. No ethical concerns were identified during the research process. The research findings are based on rigorous analysis and conducted with integrity and responsibility. Overall, this study is conducted in full compliance with ethical research practices.

