# OpenReview forum: "Uncovering the Hidden Data Costs of Mobile YouTube Video Ads"
_ACM.org/TheWebConf/2024/Conference — TheWebConf24 Oral_

### Official Review · Reviewer_rGzK · 2023-11-17

**Novelty:** 6
**Technical Quality:** 5

**Review:**

**Summary**

In this work, the authors study the effect of YouTube videos in terms of data consumption, data wastage and (hidden) monetary costs. The authors find that indeed video ads have significant impact on a user’s data plan, especially in developing countries.

**Evaluation**

This work is related to the WWW conference. The authors followed an explicit and precise methodology and found very interesting results. I have some concerns regarding the methodology and there are various things that can be improved but, generally, it is a good work.

**Originality**

Adequate. The authors perform an original study and to my understanding there isn’t similar work out there. The authors followed a very nice approach to quantify the effect of a phenomenon in a real-world scenario (i.e. developing countries). We often see researchers say tracking/ads/X/Y can have an effect on a user’s data plan but the authors went the extra step and proved it.

**Impact**

Potential. This study can have an impact both on the academic community and the real world. I don’t think the proposed solution is something that will be actually implemented since it might affect other factors. However, I think it can inspire further research on the topic. Also, this work is timely with the current controversy regarding YouTube and ad blockers. The authors might want to further discuss this in the camera-ready version.

**Citations**

I think that the authors can do a better work in the related work section and include more related works that study a similar topic. This will increase the reader’s understanding of the state-of-the-art. With only a quick search I was able to find various works that the authors did not include [1-3]. Additionally, you need a reference in Section 2.1 to support the claim “YouTube provides marketers with different ad formats including video and non video options”. Finally, reference [7] is not correct based on how it is used in section 2.2. It shows the “most popular websites in Nov 2022”, not the “top countries in terms of YT audience size”.

[1] Gui, Jiaping, et al. "Truth in advertising: The hidden cost of mobile ads for software developers." 2015 IEEE/ACM 37th IEEE International Conference on Software Engineering. Vol. 1. IEEE, 2015.

[2] Tandyonomanu, Danang. "Ads on YouTube: Skip or Watch?." 1st International Conference on Social Sciences (ICSS 2018). Atlantis Press, 2018.

[3] Belanche, Daniel, Carlos Flavián, and Alfredo Pérez-Rueda. "Brand recall of skippable vs non-skippable ads in YouTube: Readapting information and arousal to active audiences." Online Information Review 44.3 (2020): 545-562.

**Reproducibility**

This work is reproducible. The authors release their collected dataset and the source code of their data collection and processing system.

**Presentation**
Good quality of presentation. The paper is well-written. Figures and tables need some work though. Some comments I have are:
* The paragraph “First, users indirectly pay…This issue becomes even more…” is found verbatim in both the Abstract and the second paragraph of the Introduction. This is not nice. The authors should rephrase it and not simply copy-paste it.
* The Introduction section is far too long (especially for an 8 page paper). The itemized list of findings that you have in the Introduction does not fit so well here. It would be better to put them in the Conclusions section and keep only the Contributions in the Introduction. The findings can be summarized with a short paragraph in the Introduction to give a taste to the user and entice them for what is to come.
* In the first point of contributions, the link to the anonymized dataset should be a reference or a footnote, not an href.
* Figures are not very clear. The authors should provide the median value for all boxplots with a number next to each box. Also, Figures 2, 4 and 11 might be better if there is a grid on them so the reader can keep track of the values depicted.
* Table 1 is not very intuitive. I think the caption should better explain what each column represents.
* I really liked the finding in Section 3.3 that because of mid-roll ads you have to re-download aprox 1 min of the content you are trying to consume. If I were you, I would highlight it in the list of findings.

**Execution**
This study is performed in a sufficient and well-thought manner. I like that the authors show that their results are statistically significant. Some comments are:
In Introduction the authors set the narrative that internet access is extremely expensive in some countries.
* In Section 3.1 the authors mention that the average duration of a trending video is 5:15 mins long (315 secs) but later they say that an ad appears on average after 4:35 mins (4.6 mins). This does not seem correct to me. Do I understand it correctly?
* In Section 3.5, collecting Android app data is not a trivial task and there is a lot of academic work on this topic. How did you do it? More details are required here.
* The narrative you follow in this work is that video ads can have a monetary impact on users of developing countries. It would be more than interesting to translate the findings of your work into actual monetary values (i.e. dollars). I think this is missing from section 4.1. For example, you should say that the (average) 2Gb plan in country X costs Y dollars. Z of them are wasted because of ads, W because of buffer loss. You can remove the preface of each section (i.e. “In this section we…”) to save space for this analysis.

**Ethics**

The work performed in this study raises some questions. One would argue that watching 46K video ads might deplete ad budgets. I would like a short comment on this topic in the Appendix. The authors should show that either they reached out to their Institutional Review Board and got approved or that their experiments have a negligible effect on the ad ecosystem.

**Questions:**

* In Figure 7 you talk about Ad Placement but earlier on in Section 3.2 you defined the same concept as Ad Location. Do I understand this correctly? Are these concepts the same? If not, you should clarify.
* There is anecdotal evidence that YouTube does not show any ads in developing countries (e.g. Ethiopia, Myanmar, Moldova). Have you looked into this? What is the reasoning behind this decision? How is a user of such a country favored compared to users of the developing countries you studied? How much data do they save? It would be interesting to see the difference in data consumption when watching a video in Myanmar vs Pakistan for example.
* In Section 2.2, could there be an overlap between the two datasets of collected videos? Could it be that a trending video also appears in the home page? Have you checked that?
* In Section 2.3, how did you emulate a mobile device? Did you use an emulator? Did you simply modify the browser’s viewport? I would expect that a different approach here might yield different results.
* I am concerned regarding the methodology presented in the “Data Collection” segment of Section 2.3. How do you measure consumed video/ad data? Do you only extract it from the “Stats for Neers” tab? If so, do you have an idea of how accurate they are? Does YT provide exact data or are they simply estimations? Do you have access to documentation on how they are computed? For example, do they show simply video data or do they also include HTTP headers, etc. This could potentially affect your findings but I guess what you found is simply a lower limit. Altogether, I would like some more text there to explain the details of your methodology.
* In Section 3.1 is the mentioned reduction for the same video?
* In Section 3.4 the assumption that ads are skipped after 5 secs sounds realistic and correct to me. Do you have any empirical data on how often this happens? Are there any estimations on what % of users actually skip ads?
* In Section 3.6, are you somehow able to set the ad quality/resolution yourselves? Is 360p a YouTube default or did you select it?

**Ethics Review Description:**

Possibly no issues.

**Reviewer Confidence:**

3: The reviewer is confident but not certain that the evaluation is correct

**Scope:**

4: The work is relevant to the Web and to the track, and is of broad interest to the community

---

### Official Review · Reviewer_KHZB · 2023-11-23

**Novelty:** 3
**Technical Quality:** 4

**Review:**

The paper investigates the bandwidth overhead of video ads on YouTube.  The paper tries to make the point that the overhead of video ads in developing countries is prohibitive for users.

For this, the paper sets up a measurement framework based on selenium that watches trending videos and records the bandwidth used by both the main video as well as video ads shown while streaming. The paper finds that video ads incur a 13.2% overhead. The paper also performs a deep dive in the overhead of different ads (skipable, mid-video) and the overhead of buffering of both main videos and video ads.

The paper has a fundamental flaw in the measurement setup:  the paper is doing all the measurements from the same vantage point — the paper does not actually perform the measurement from vantage points in the different countries it considers (4 developed and 4 developing). This is very problematic as most ads are targeted to particular countries, and video ads delivered in one country are probably not representative of video ads in other countries. It is, for example, possible that developed countries receive more ads on YouTube than developing countries; hence, they incur higher bandwidth overheads.

I am also not convinced about the problem tackled; if mobile data is very expensive in some countries, I would assume users will simply not go on YouTube and will wait to have wifi access to watch videos on YouTube.

Finally, the paper proposes solutions for buffering that might be of interest.

After rebuttal:
- the authors clarified that they performed the analysis from different vantage points, so this alleviates the biggest measurement flaw I pointed out.

**Questions:**

How would the results change if the vantage points would be in the different countries studied ?

**Reviewer Confidence:**

4: The reviewer is certain that the evaluation is correct and very familiar with the relevant literature

**Scope:**

3: The work is somewhat relevant to the Web and to the track, and is of narrow interest to a sub-community

---

### Official Review · Reviewer_e6xH · 2023-11-23

**Novelty:** 6
**Technical Quality:** 6

**Review:**

The paper describes a measurement of data use and loss associated with YouTube ads. The authors underscore the uneven impact of that loss across populations, since in developing countries, mobile data is particularly expensive as a fraction of income.

The authors present interesting findings, for example: skipped ads are mostly pre-buffered, so skipping does not save all of the data; ads played in the middle of the video cause the loss of pre-buffered actual video.

The procedure is clearly described, the importance of the problem is well argued, and the authors propose potential solutions and simulate their efficiency. The authors measure both the native YouTube mobile app and the m.youtube.com website.

In general, I enjoyed reading the paper. I only have a few the minor remarks:
- while the authors claim to have analyzed the mobile app, there are no actual results in the paper
- the loss reductions should be expressed in terms of percentage - what is a 1.2x reduction? At first I thought it's a reduction by 20%, but then another reduction is reported to be 5.6x. What would that mean?
- the figures should have a grid for easier comparison on values
- If the paper is not accepted to WWW, I suggest the authors try the Internet Measurement Conference where this paper might be a better fit.

**Questions:**

I would implore the authors to at least include some summary statistics on from the measurement of the mobile app. As it is right now, we have to take their word for it.

**Reviewer Confidence:**

2: The reviewer is willing to defend the evaluation, but it is likely that the reviewer did not understand parts of the paper

**Scope:**

3: The work is somewhat relevant to the Web and to the track, and is of narrow interest to a sub-community

---

### Official Review · Reviewer_xPv8 · 2023-11-24

**Novelty:** 5
**Technical Quality:** 6

**Review:**

In this paper, the authors examine the hidden costs of YouTube ads, as measured in terms of wasted bandwidth. The authors identify this as an equity issue, especially in the developing world where smartphones are the primary way to use the Internet, and data plans are far more expensive. The authors identify surprising bugs in YouTube that waste valuable bandwidth.

I confess, I expected to not like this paper: the topic is quite narrow. But I ended up liking the paper quite a bit! While the topic is narrow, the analysis is very thorough, and the amount of waste identified by the authors surprised me. Given YouTube's popularity, I agree that this waste could be an equity problem, and I hope that by publishing this study the authors will encourage YouTube to make engineering changes to their platform.

I have some methodological questions as well as presentation suggestions:

3.1: Here, it would be useful to add a table of summary statistics such as: median video length, number of ads of different types (skippable/non-skippable, pre, mid, and post), median ads per video, median ad lengths, etc., broken down by trending and non-trending, and possibly also by developed and non-developed country.

3.1: I'm surprised that the Ad Gap is smaller in developing nations. I would assume that developed nations have more sophisticated companies with larger marketing budgets, and thus there would be more ads in developed nations. What explains this inversion?

3.2, line 449: The authors refer to a highlight in Figure 1 but I'm not able to find this highlight.

3.2, line 485: Here the authors speculate about what users are likely to experience. I worry that this speculation is too speculative: to claim that users experience specific data loss conditions assumes that YouTube users view trending and other videos recommended on the YouTube homepage, since that is the data being analyzed here. It is unclear what the data loss trends would be if other YouTube usage patterns had been evaluated by the crawler, such as following up-next recommendations, or watching videos from particular categories exclusively. I think the underlying issue is that YouTube has an incentive to recommend videos (e.g., on the homepage) that are monetizable, but not all videos on YouTube are monetizable, so it may be the case that data loss patterns are less, or at least different, under ecological viewing conditions.

3.4, line 595: It is not good practice to ask the reader to compare two figures that are on different pages. If this comparison is important (and I think it is, in this paper), the authors should seriously consider combining Figures 1 and 8. There is lots of space around Figure 1 so this seems possible.

Figure 12(a) and (b) should really be a single figure with one x-axis. At a minimum, the two y-axes need to be made identical so the bar heights are comparable across the two subfigures.

4.2: For me, this was the least compelling section of the paper. First, as the authors note, moving to double ads may have a negative impact on user experience. Indeed, I suspect YouTube has tried this approach, observed negative impact, and moved away from this approach. Second, maybe I'm being naive, but why isn't the solution to simply not throw away the main-video buffer when a mid-roll ad plays? Why can't their be a main-video buffer and an ad buffer? The authors suggest this solution, and others, in Section 5, and they are far superior to the approach evaluated in 4.3.

**Questions:**

2.3: Did the authors check to make sure all VPN endpoints were properly geolocated within the eight target countries?

2.3: How did the scrapers manage cookies? I ask because the YouTube homepage is personalized, so maintaining cookies over time may cause the mix of videos on the homepage to shift.

2.3: Did the authors configure the language settings of their crawler to match that of the local language in each country? For example, was the language of the crawler set to Japanese when the crawler was using the VPN to download data from Japan?

2.3: Did the authors do any manual checking to make sure the ads they were receiving in different locations correctly matched each location? For example, were the ads served in Japan for Japanese companies?

**Reviewer Confidence:**

3: The reviewer is confident but not certain that the evaluation is correct

**Scope:**

3: The work is somewhat relevant to the Web and to the track, and is of narrow interest to a sub-community

---

### Official Review · Reviewer_4sQ1 · 2023-12-01

**Novelty:** 4
**Technical Quality:** 2

**Review:**

The paper aims to characterize hidden data costs associated with Youtube ads in mobile platforms. The paper starts from tens of thousands of videos and respective ads collected in 8 countries and evaluated them in terms of the impact of the ads in the bandwidth consumption. The motivation of the paper is that such ads affect significantly less wealthy countries, where connectivity costs are higher.

I believe the analysis has a fundamental caveat by not considering the watching behavior of users and how these behaviors change as a function of bandwidth availability and connectivity costs. For instance, in some countries there is a significant fraction of pre-paid phones. Further, the percentage of accesses from wifi networks (at home or at work) should be factored out for the analysis proposed. Other examples of issues are whether popular videos do contain more ads or ads that last longer and the percentage of the video that is effectively watched by users, among others.

I also suggest the authors check other information such as the number and characteristics of the phones and other mobile platforms in each country. It may give you more detailed information about the equipment profile and the typical user investment for accessing Internet content.

Regarding the affordability analysis, I got really curious about Brazil, which is not a rich country, but presents numbers significantly different from the other countries. Again, the discussion is pretty much descriptive and does not provide a deeper understanding of the reasons behind the numbers.

In summary, although the topic is relevant and challenging, I believe that the analysis performed is not mature enough, since it does not consider both user behavior characteristics and contextual features on the various countries.

**Questions:**

Can you explain Brazil numbers in the affordability analysis?

Did you consider the user behavior uniform across countries?

It is not clear to me whether the differences are really as significant as the authors state. For instance, some of the boxplots (e.g., Figures 6 and 7) show quite an overlap among scenarios.

**Reviewer Confidence:**

3: The reviewer is confident but not certain that the evaluation is correct

**Scope:**

3: The work is somewhat relevant to the Web and to the track, and is of narrow interest to a sub-community

---

### Decision · Program_Chairs · 2024-01-22

**Decision:**

Accept (Oral)

**Comment:**

We support the area chair's recommendation  (below) to accept this paper. We ask the authors to improve the manuscript for camera-ready based on reviewers' feedback. We particularly underline the ethics flag raised by reviewer rGzK and ask the authors to improve the work, as they communciated in their response to the reviewer, for camera-ready.

"The authors present an empirical analysis of the data costs of mobile YouTube video ads, particularly the impact of users in developing countries. The work is generally clear and highlights important equity issues in data consumption through video ads. Reviewers raised concerns about the depth/maturity of the analysis and presentation. However, the authors show a willingness to address the reviewers' concerns, and I recommend acceptance."